# Hf Doping Boosts the Excellent Activity and Durability of Fe-N-C Catalysts for Oxygen Reduction Reaction and Li-O_2_ Batteries

**DOI:** 10.3390/nano14242003

**Published:** 2024-12-13

**Authors:** Mingrui Liu, Shaoqiu Ke, Chuangqing Sun, Chenzhuo Zhang, Shijun Liao

**Affiliations:** 1National Energy Key Laboratory for New Hydrogen-Ammonia Energy Technologies, Foshan Xianhu Laboratory, Foshan 528200, China; 2Hubei Key Laboratory of Photoelectric Materials and Devices, School of Materials Science and Engineering, Hubei Normal University, Huangshi 435002, China; 3State Key Laboratory of Advanced Technology for Materials Synthesis and Processing, Wuhan University of Technology, Wuhan 430070, China; scq11@whut.edu.cn (C.S.); 18271774069@163.com (C.Z.); 4The Key Laboratory of Fuel Cell Technology of Guangdong Province, School of Chemistry and Chemical Engineering, South China University of Technology, Guangzhou 510641, China; chsjliao@scut.edu.cn

**Keywords:** non-noble metal catalyst, oxygen reduction reaction, active sites, durability, PEMFCs, Li-O_2_ batteries

## Abstract

Developing highly active and durable non-noble metal catalysts is crucial for energy conversion and storage, especially for proton exchange membrane fuel cells (PEMFCs) and lithium-oxygen (Li-O_2_) batteries. Non-noble metal catalysts are considered the greatest potential candidates to replace noble metal catalysts in PEMFCs and Li-O_2_ batteries. Herein, we propose a novel type of non-noble metal catalyst (Fe-Hf/N/C) doped with Hf into a mesoporous carbon material derived from Hf-ZIF-8 and co-doping with Fe and N, which greatly enhanced the activity and durability of the catalyst. When applied in the cathode of PEMFCs, the current density can reach up 1.1 and 1.7 A cm^−2^ at 0.7 and 0.6 V, respectively, with a maximum power density of 1.15 W cm^−2^. The discharge capacity of the Li-O_2_ batteries is up to 15,081 mAh g^−1^ with Fe-Hf/N/C in the cathode, which also shows a lower charge overpotential, 200 mV lower than that of the Fe/N/C. Additionally, the Fe-Hf/N/C catalyst has demonstrated better stability in both PEMFCs and Li-O_2_ batteries. This reveals that Hf can not only optimize the electronic structure of iron sites and increase the active sites for the oxygen reduction reaction, but can also anchor the active sites, enhancing the durability of the catalyst. This study provides a new strategy for the development of high-performance and durable catalysts for PEMFCs and Li-O_2_ batteries.

## 1. Introduction

The growing demand for sustainable energy solutions has prompted significant research into advanced materials for electrochemical applications, particularly in fuel cells and metal–air batteries [1,2,3]. The oxygen reduction reaction (ORR) and oxygen evolution reaction (OER) are crucial processes for proton exchange membrane fuel cells (PEMFCs) and metal–air batteries, which directly determine the output performance of the overall system [4,5]. However, the sluggish reaction kinetics result in poor efficiency, insufficient durability, and low-rate capability, which severely limit their commercial development [6,7]. Although traditional catalysts, such as platinum and its alloys, are effective, they are often hindered by high costs and resource scarcity. Consequently, the exploration of alternative materials has become increasingly important.

Recent advancements [8,9,10] have highlighted the potential of dual-atom catalysts as a new class of catalysts for oxygen electrocatalytic reactions. Bimetallic systems, consisting of two different metal atoms, can leverage the synergistic effects of their unique electronic and geometric properties. When these metals are arranged at the atomic level, they can enhance catalytic activity and improve selectivity. In the context of PEMFCs, dual-atom catalysts can optimize the adsorption and activation of oxygen species, which is crucial for improving efficiency in fuel cells. The incorporation of a secondary metal into the active site can modulate the electronic structure and mitigate carbon and active site oxidation, thus altering the adsorption energies of reaction intermediates and promoting more efficient electron transfer, ultimately leading to more excellent activity and durability at lower costs compared to traditional catalysts [11,12,13]. Simultaneously, the development of Li-O_2_ batteries also needs efficient catalysts to accelerate the reduction and evolution of oxygen during charge and discharge cycles [14,15]. Dual-atom catalysts hold promise in this area by potentially adjusting the formation/decomposition process of discharge products and oxidation overpotentials, thereby enhancing the overall energy efficiency and cycle stability of Li-O_2_ batteries [16,17,18]. In particular, for non-noble metal catalysts, such as Fe, Ni and Co, perovskite, and oxides, induced substrate effects and synergistic effects are beneficial to ORR/OER [19,20,21,22]. Soltani et al. [19] discovered that the interaction between perovskite and the substrate can modify the electronic structure and oxygen adsorption/desorption kinetics, thereby enhancing the ORR/OER activity. Amin et al. [20] showed that the combination of different metals can lead to enhanced catalytic activity. Li et al. [23] investigated the mild spin magnetization and finite spin-polarized conduction electrons of Fe-Ni diatomic sites for ORR/OER by DFT calculations and experimental tests, demonstrating that the formation of Fe-Ni sites facilitates the absorption and desorption pathways of O_2_, and then enhances the catalytic activity of ORR/OER. He et al. [24] introduced Fe^3+^ and Co^2+^ ions into the pyrolytic ZIF-8 carbon framework to obtain an Fe_1_Co_3_-NC-1100 dual-atom catalyst, which showed superior activity and an excellent kinetic process compared to commercial 40 w% Pt/C. Dual-atom catalysts like FeZn [25], FeSe [26], NiCo [27], etc., have been studied successively for optimizing ORR activity or OER activity, and they exhibit comparable or even better electrochemical performance than that of commercial catalysts in the rotating disk electrode. However, their performance in PEMFCs still needs to be improved, and their application in Li-O_2_ batteries has rarely been studied. Therefore, exploring and understanding of the reaction mechanism of dual-atom catalysts in electrocatalytic devices is of great significance for future research.

In this study, a mesoporous carbon material derived from Hf-ZIF-8 doped with Hf was used as the support to prepare a carbon-based catalyst Fe-Hf/N/C co-doped with the pre-transition metal Hf atom, the post-transition metal Fe atom, and N. Because of introducing the Hf atom, the catalytic activity and durability of the catalyst have both been significantly improved. The Fe-Hf/N/C catalyst with dual-atom dispersion not only had improved performance in PEMFCs but also had enhanced stability/durability in Li-O_2_ batteries. And the double monatomic catalyst was used in a Li-O_2_ battery for the first time, which increased the battery capacity and increased the cycle life. The Fe-Hf/N/C catalyst achieved superhigh current densities of 1.1 and 1.7 A cm^−2^ at voltages of 0.7 V and 0.6 V, respectively, and a maximum power density of 1.15 W cm^−2^ in PEMFCs. The stability of the catalyst is also significantly improved compared with the Fe/N/C catalyst. This study provides a new strategy for the development of high-performance and durable catalysts for both PEMFCs and Li-O_2_ batteries.

## 2. Experimental Section

### Sample Preparation

Preparation of precursor ZIF-8 and ZIF-8-Hf: The experimental procedure is described in ESI, S1. A series of Hf-ZIF-8 precursors with different Zn/Hf mass ratios were named as Hf-ZIF-8-x/y, where x/y is the mass ratio of Hf to Zn.

Synthesis of catalysts: The Fe-Hf/N/C and Fe/N/C catalysts were synthesized via a high-temperature pyrolysis method combined with a gaseous doping approach. The experimental procedure is detailed in ESI, S1. The obtained catalysts are named as Fe-Hf/N/C-x/y-a/b-T, where x/y represents the mass ratio of hafnium acetylacetonate in the precursors, a/b represents the mass ratio of precursors to ferrocene, and T represents the pyrolysis temperature.

Measurements of MEAs and Li-O_2_ battery: The experimental procedure, characterization, property measurement, and single proton exchange membrane (PEM) fuel cell and Li-O_2_ battery testing methods are described in ESI, S1.

## 3. Results and Discussion

### 3.1. Structure Characterization

To determine the effect of doping with different mass ratios of zinc nitrate hexahydrate/hafnium acetylacetone (Zn/Hf) on the structure of precursor ZIF-8, the XRD patterns of ZIF-8 doped with different mass ratios of Zn/Hf were tested (Figure 1a). The XRD patterns of ZIF-8 were fitted through its cif file; the corresponding PDF card number is [PDF# 00-062-1030]. It can be seen that the structure of the precursors obtained by doping with different Zn/Hf mass ratios has almost no change, and the crystal phase structure is better maintained. However, with the gradual increase in Hf mass, the peak intensity of the ZIF-8 precursors with different Zn/Hf mass ratios gradually decreases. In particular, when the mass ratio of Zn/Hf is 3/400 and 4/400, the diffraction peak intensity is significantly weakened. This result indicates that excessive Hf doping in ZIF-8 will not change the crystal phase structure, but will reduce the crystallinity of the sample.

To investigate the microstructures of Fe-Hf/N/C catalysts, low- and high-magnification SEM images were captured and are displayed in Figure 2a,b. It can be seen that the structured polyhedral morphology of Fe-Hf/N/C-2/400 was damaged to some extent. Although the polyhedral morphology is basically maintained, the surface is rough and local areas appear concave. This topography is beneficial for the dispersion of the catalyst active site on the catalyst surface and the accessibility between oxygen and the active site during the ORR process. To further investigate the roles of Hf in Fe-Hf/N/C, Figure 2c shows the HAADF-STEM image of Fe-Hf/N/C-2/400 and corresponding elemental mappings of Fe, Hf, C, and N. The Fe-Hf/N/C-2/400 can still maintain a good polyhedral structure after secondary high-temperature pyrolysis. The Fe, Hf, N, and C elements in the catalyst are uniformly dispersed in the catalyst skeleton. Moreover, there are no metal particles on the skeleton or surface of the Fe-Hf/N/C catalyst, suggesting that Fe and Hf are likely coordinated with N and dispersed atomically within the catalyst framework.

To further clarify the influence of doping with Zn/Hf on the precursor ZIF-8 structure, the structures of ZIF-8 and Hf-ZIF-8-2/400 (the precursor to synthesize the optimal catalyst) were analyzed by FT-IR. The characteristic peaks in FT-IR spectra of ZIF-8 and Hf-ZIF-8-2/400 are very consistent in Figure 1b, which indicates that there is no coordination to form new groups after Hf doping into ZIF-8 [28]. Combined with the results of XRD patterns, it is further proved that the introduction of Hf does not change the original structure of ZIF-8. Moreover, the Hf is likely to fill the pores [4,29,30] of ZIF-8 during the synthesis process of the precursor Hf-ZIF-8-2/400.

To obtain the catalyst with stable crystal structure and excellent catalytic performance, the ZIF-8 precursors and a series of Hf-ZIF-8 precursors were pyrolyzed at a high temperature combined with Fe doping. The XRD patterns of the obtained samples are presented in Figure 1c. The characteristic peaks (002) and (004) corresponding to carbon are evident in all precursors after catalyst pickling and secondary pyrolysis, while the characteristic peaks of Fe or Hf compounds do not appear. As a result, the Fe and Hf in the Fe-Hf/N/C catalysts did not exist as a substance or compound, possibly atomically dispersed in the catalyst.

According to the BET test results (ESI, S2), the introduction of Hf can improve the specific surface area of the Fe/N/C catalyst (994.37 m^2^ g^−1^), facilitating the dispersion of active sites on the catalyst surface, and provide more surface area for the adsorption of active species and the deposition of battery products. This reduces performance degradation and extends the durability of the electrocatalytic device.

To investigate the influence of Hf doping on the elemental composition and electronic structure of the Fe/N/C catalysts, the Fe/Hf/N/C and Fe/N/C catalysts were characterized by XPS. The high-resolution XPS spectra and the corresponding peak fitting results of Fe 2p and N1s on Fe-Hf/N/C and Fe/N/C catalysts are shown in Figure 3 and Appendix A. Compared with the Fe/N/C catalysts, the binding energy of both the Fe and N in Fe-Hf/N/C catalysts decreases, while the binding energy of Hf increases compared to that of mono-metal Hf/N/C [31]. It is proven that there is an obvious electronic interaction between Fe and Hf in Fe-Hf/N/C, and some electrons of Hf are transferred to Fe, cooperatively optimizing the activity and stability of Fe active sites [30,32]. The types and contents of nitrogen species in Fe-Hf/N/C and Fe/N/C catalysts were analyzed according to the N 1s high-resolution spectra and peak fitting results (Figure 3d). The Fe-Hf/N/C catalyst contains graphitic N, pyridine N, oxidized N, pyrrole N, Fe-N, and Hf-N species. It has been reported that Fe-N and Hf-N species exhibit ORR catalytic activity [33,34]. At the same time, the content of graphitic N and pyridine N in the Fe-Hf/N/C catalyst is significantly higher than that in the Fe/N/C catalyst, while the content of oxidized N and pyrrole N is lower than that in the Fe/N/C catalyst. High content of graphitic N and pyridine N is also beneficial for the conductivity of the catalyst, enhancing its intrinsic catalytic activity [35,36]. Consequently, the introduction of Hf into the Fe/N/C catalyst can improve the activity of the ORR.

The chemical state and coordination environment of Fe and Hf in the Fe-Hf/N/C catalyst were characterized by X-ray absorption near-side structure (XANES) and extended X-ray absorption fine structure (EXAFS), and the results are shown in Figure 4 and Appendix A. The critical value of Fe K-edge energy absorption of Fe/N/C and Fe-Hf/N/C is between the Fe foil and Fe_2_O_3_, and close to Fe_2_O_3_ (Figure 4a), which indicates that the chemical state of Fe in Fe/N/C and Fe-Hf/N/C is between Fe^0^ and Fe^3+^ and nearer to Fe^3+^ [37,38]. Figure 4b is the Fourier transform (FT) K^3^-weighted χ(k)-function curve (R-space) of the Fe k-edge EXAFS spectrum. It can be seen that Fe/N/C and Fe-Hf/N/C have a main peak at 1.40 Å, which belongs to the main Fe-N coordination [30,39,40], while Fe/N/C and Fe-Hf /N/C spectra do not show Fe-Fe peaks, indicating that Fe in the catalyst exists in the form of atoms and coordinates with N to form an Fe-N active center. Moreover, the average coordination number of Fe-N can be obtained as 4.7 (ESI, S3), suggesting that Fe-N coordination in the catalyst is mainly a mixture of Fe-N_4_ and Fe-N_5_.

### 3.2. Excellent Electrochemical Properties of Catalysts

The ORR performance of the catalyst was assessed using a rotating disk electrode (RDE) in O_2_-saturated 0.1 M HCIO_4_ solution. The LSV curves of the N/C, Pt/C, Hf/N/C, Fe/N/C, and Fe-Hf /N/C catalysts are presented in Figure 5a. The half-wave potential of Pt/C is 0.87 V (vs. RHE); that of the Fe-Hf/N/C catalyst is up to 0.84 V (vs. RHE), which is 80 mV higher than that of the Fe/N/C catalyst (0.76 V vs. RHE). Although Pt/C shows relatively high catalytic activity, our Fe-Hf/N/C catalyst exhibits a comparable half-wave potential and even demonstrates advantages in terms of durability and cost-effectiveness. Meanwhile, this indicates that the intrinsic catalytic activity of the catalyst can be enhanced by Hf co-doping. In addition, in order to maximize the excellent catalytic activity of Fe-Hf/N/C catalysts, we further optimized the preparation process of the catalyst. The electrochemical performance of catalysts prepared by Zn/Hf precursor systems with different mass ratios are shown in Figure 5b. It reveals that with the increase in the mass of Hf in the precursor, the catalytic activity of the catalyst increases first and then decreases. The half-wave potential of catalyst is above 0.8 V when the mass ratio of Hf/Zn is 2/400. Additionally, the pyrolysis temperature of the Fe-Hf/N/C catalyst also has an effect on the catalytic performance. As the temperature rises, the current density increases, but the half-wave potential increases first before declining, with the most positive potential observed at 900 °C (Figure 5c). Furthermore, the mass ratio of Hf-ZIF-8 and ferrocene was also investigated (Figure 5d); the results show that when the mass ratio is 9:13, the catalyst reaches the best catalytic activity.

To reveal the electrochemical active surface area (ECSA) of the catalysts Fe-Hf/N/C and Fe/N/C, the CV curves of different sweep velocities in the non-Faraday-effect region were tested. It was shown that the current density of both Fe-Hf/N/C and Fe/N/C electrodes increases with higher scanning speed (Figure 6a,b). This is attributed to the fact that the high scanning rate promotes the electron transfer reaction rate on the surface of the electrode material. Moreover, the ECSA is a measurement of the contribution of electrode surface area to electrochemical activity; the ECSA of Fe-Hf/N/C and Fe/N/C electrodes was calculated under various sweep rates, as shown in Figure 6c,d. The Fe-Hf/N/C electrode provided a promising ECSA of 54.70 mF cm^−2^ at the scan rate of 5–25 mV s^−1^ in the three-electrode test, while the Fe/N/C electrode resulted in an ECSA of 14.98 mF cm^−2^ at the same scan rate. This significant difference is primarily due to the incorporation of Hf and Fe atoms in the Fe-Hf/N/C electrode, which increases the active sites on the electrode surface, thus enhancing the electron transfer and electrochemical activity.

### 3.3. Application in Proton Exchange Membrane Fuel Cells

To further characterize the performance of the catalysts in PEMFCs, Fe-Hf/N/C catalysts with different loads were used as cathode catalysts for a membrane electrode assembly (MEA), and their performance was tested on the Arbin fuel cell test system, as shown in Figure 7. It can be seen that the voltage, power density, and maximum power density of the catalyst Fe-Hf/N/C increase first and then decrease with increasing loading (Figure 7a). The catalyst achieves the highest power density when the Fe-Hf/N/C loading is 1.5 mg, showing outstanding MEA performance. Under H_2_-O_2_ conditions, the current densities of Fe-Hf/N/C at 0.7 and 0.6 V are 1.1 and 1.7 A cm^−2^, respectively, and the maximum power density is 1.15 W cm^−2^. Additionally, the Fe-Hf/N/C catalyst exhibited superior durability in an actual single PEM fuel cell (Figure 7b). When the cell was supplied with hydrogen and pure oxygen operating at a current of 1.0 mA/cm^2^ for 20 h, the voltage of the MEA with an Fe-Hf/N/C cathode reduced by only 23.4%, much smaller than the 72.4% attenuation of the Fe/N/C cathode. This phenomenon is mainly due to the co-doping of Hf and Fe atoms, which can increase the superior acid tolerance of Fe-Hf/N/C [34], thereby promoting durability in an acidic environment.

### 3.4. Application in Li-O_2_ Battery

Understanding the impact of catalysts on Li-O_2_ battery performance is essential for developing advanced energy storage devices. To evaluate the role of catalysts in the Li-O_2_ battery field, the charge and discharge performance and cycle stability of Fe-Hf/N/C and Fe/N/C cathode catalysts were investigated, as shown in Figure 8. Figure 8a is a schematic diagram of the Li-O_2_ battery test device. Generally, the ORR occurs during the discharge phase, where O_2_ is reduced at the cathode, and leads to the formation of lithium peroxide (Li_2_O_2_) or other lithium oxides, while the OER takes place during the charging phase, where the lithium oxides are oxidized back to oxygen. Additionally, considering that the properties of the electrolyte solvent can also affect the oxygen reduction and oxygen evolution performance of Li-O_2_ batteries [41], the same electrolyte was used for all test samples. Figure 8b presents the cyclic voltammetry of Fe/N/C and Fe-Hf/N/C catalysts in a non-aqueous Li-O_2_ battery. Fe-Hf/N/C exhibited a higher peak current density, more positive onset potential, and lower onset OER potential than the Fe/N/C catalyst. The excellent electrocatalytic activity is attributable to the high surface area and high density of active centers caused by the atomically dispersed Fe and Hf, which effectively promote the reaction kinetics and thereby reduce the charge overpotential. The rate capabilities of Fe/N/C and Fe-Hf/N/C catalysts are depicted in Figure 8c. The battery with the Fe-Hf/N/C cathode delivered a discharge capacity of up to 15,081 mAh g^−1^, which was nearly two times higher than that of the Fe-/N/C cathode at 100 mAg^−1^. The Fe-Hf/N/C cathode also showed a lower charge overpotential, which was 200 mV lower than that of the Fe/N/C. It is noteworthy that with the increase in current density, the advantages of the Fe-Hf/N/C catalyst can still be better maintained, verifying its excellent ORR/OER activities. The battery cycle stability test was carried out under a limited capacity of 1000 mAh g^−1^ and at a rate of 200 mA g^−1^. The Fe-Hf/N/C cathode exhibited far better cycling stability over 200 cycles with terminal charge voltages always below 4.0 V (Figure 8e). In comparison, the battery with the Fe/N/C cathode failed after about 30 cycles (Figure 8f). This indicates that the inclusion of Hf elements in the catalyst structure can enhance the electrochemical performance by stabilizing the operating voltage, which is crucial for maintaining efficient energy conversion in Li-O_2_ batteries. The differences in performance between Fe-N/C and Fe-Hf/N/C can be attributed to the presence of Hf atoms in the latter, which likely contribute to enhanced electronic conductivity and catalytic activity. These effects improve the overall electrochemical reaction rates and reduce overpotential, leading to better battery performance and significantly improved cycling stability.

## 4. Conclusions

In this study, a series of Fe-Hf/N/C catalysts with different Zn/Hf mass ratios were synthesized by a high-temperature pyrolysis method and gaseous doping method. The phase constituents and morphologies of the catalysts were characterized by various methods. The results demonstrated that the Fe in the Fe-Hf/N/C catalyst was atomically dispersed, and no chemical reaction occurred between the Fe and Hf elements. The electrochemical properties of the catalysts were tested and attested that the Fe-Hf/N/C catalysts possessed high oxygen reduction reaction (ORR) activity and stability. The introduction of Hf into the Fe/N/C catalyst can significantly enhance the activity of the ORR, as well as the durability of PEMFCs and Li-O_2_ batteries. The enhancement can be ascribed to possible synergistic effects of Hf and Fe atoms, which optimized the activity and acid tolerance of Fe active sites. The findings of this study suggest that Fe-Hf/N/C catalysts hold great potential for application in PEM fuel cells and Li-O_2_ batteries.

## Figures and Tables

**Figure 1 nanomaterials-14-02003-f001:**
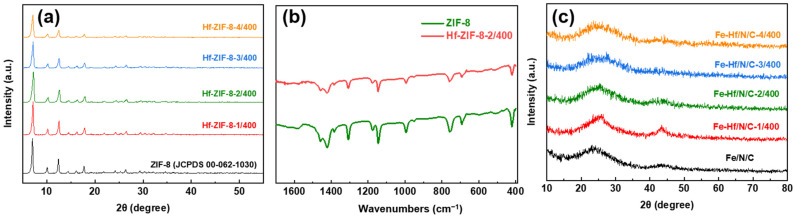
Phase constituents: (**a**) XRD patterns of precursors ZIF-8 and ZIF-8 with different Zn/Hf mass ratios. (**b**) FT-IR spectra of precursors ZIF-8 and Hf-ZIF-8-2/400. (**c**) XRD pattern of the Fe/N/C and Fe-Hf/N/C catalysts with different Zn/Hf mass ratios.

**Figure 2 nanomaterials-14-02003-f002:**
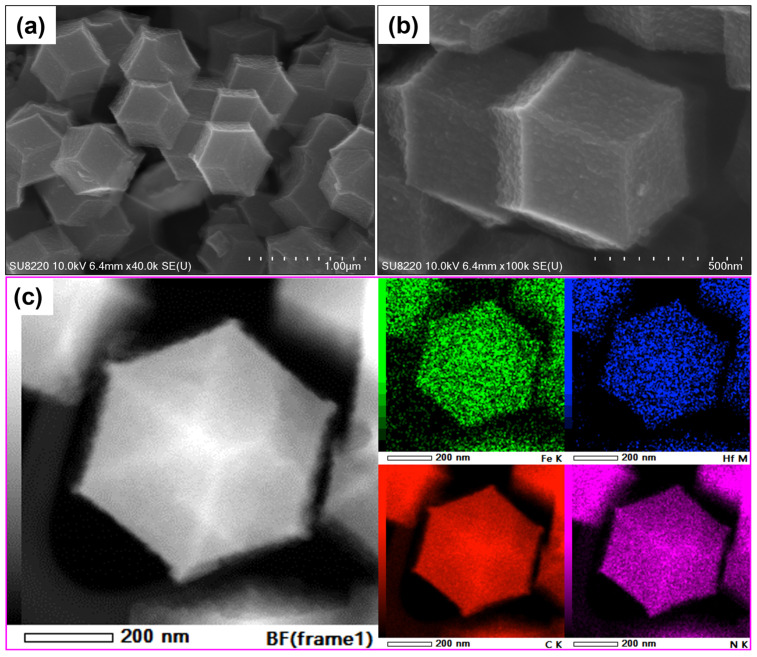
Microstructures: (**a**) Low- and (**b**) high-magnification SEM images of Fe-Hf/N/C-2/400. (**c**) HAADF-STEM of Fe-Hf/N/C-2/400 and corresponding elemental mappings of Fe, Hf, C, and N.

**Figure 3 nanomaterials-14-02003-f003:**
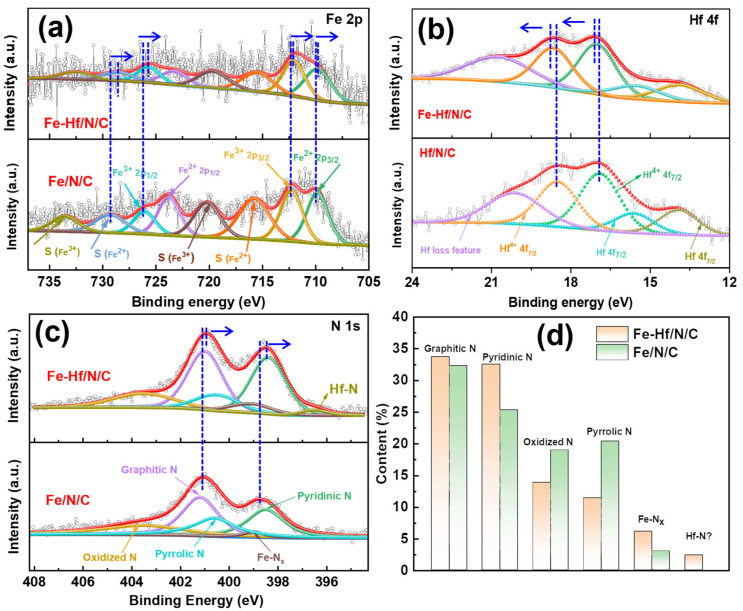
Electron structure. High-resolution (**a**) Fe 2p, (**c**) N1s, XPS spectra of Fe-Hf/N/C and Fe/N/C. (**b**) High-resolution Hf 4f XPS spectra of Fe-Hf/N/C and Hf/N/C. (**d**) Content of different N species for Fe-Hf/N/C and Fe/N/C.

**Figure 4 nanomaterials-14-02003-f004:**
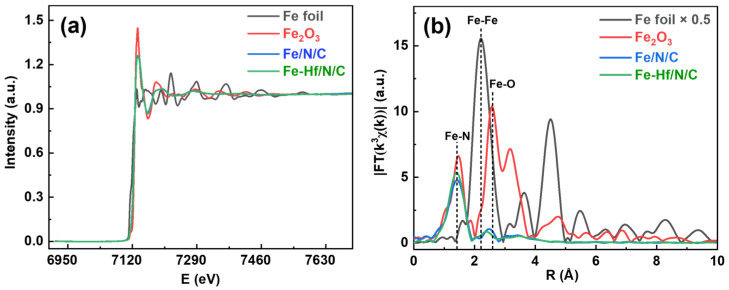
XAFS analysis. Fe K-edge XANES spectra (**a**) and Fourier transform of Fe K-edge EXAFS spectra (**b**) for Fe/N/C, Fe-Hf/N/C, and reference Fe foil, Fe_2_O_3_.

**Figure 5 nanomaterials-14-02003-f005:**
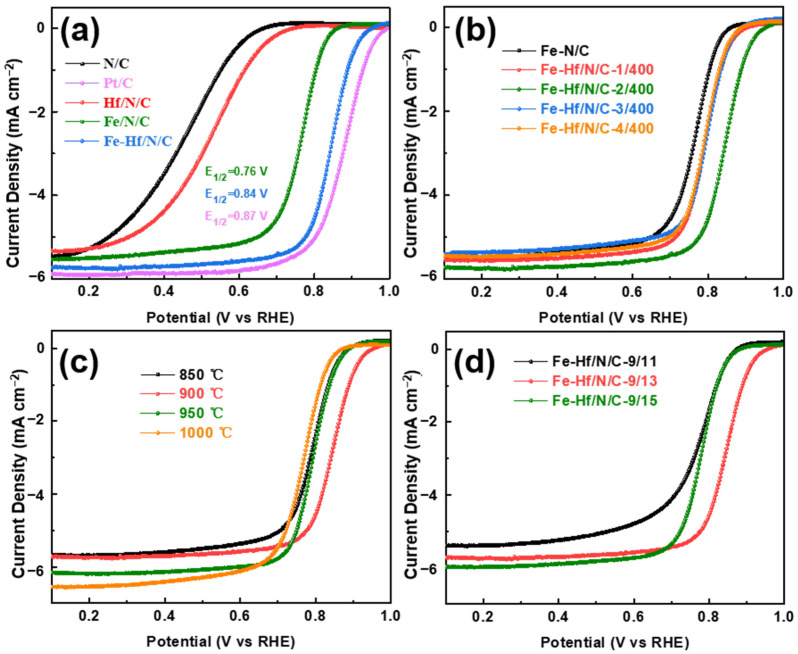
Electrochemical properties: The LSV curves of (**a**) the N/C, Pt/C, Hf/N/C, Fe/N/C, and Fe-Hf/N/C; (**b**) the Fe-Hf/N/C catalysts with different amounts of Zn/Hf; (**c**) the Fe-Hf/N/C catalysts pyrolyzing under different temperatures; and (**d**) the Fe-Hf/N/C catalysts with different amounts of Hf-ZIF/Fe.

**Figure 6 nanomaterials-14-02003-f006:**
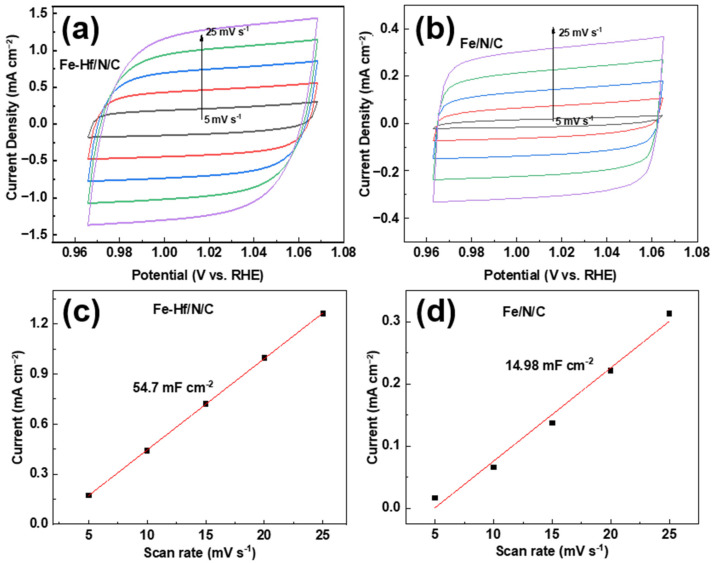
Electrochemically active surface area: The CV curves for (**a**) Fe-Hf/N/C and (**b**) Fe/N/C electrodes under various sweep rates. The ECSA of (**c**) Fe-Hf/N/C and (**d**) Fe/N/C electrodes under various sweep rates.

**Figure 7 nanomaterials-14-02003-f007:**
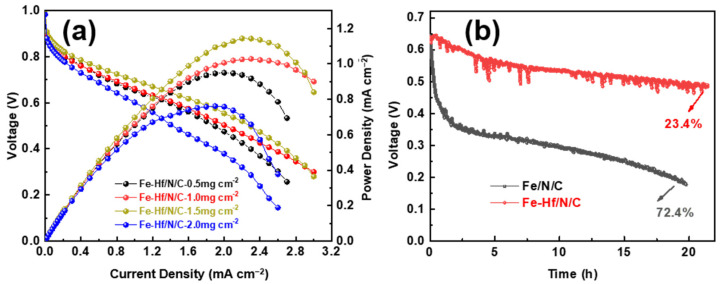
Hydrogen oxygenation properties: (**a**) Single PEM fuel cell performance of Fe-Hf/N/C with different loads as cathode catalyst under conditions of H_2_-O_2_. (**b**) Stability/durability of PEM single-cell performance of Fe-Hf/N/C and Fe/N/C as cathode catalyst under H_2_-O_2_.

**Figure 8 nanomaterials-14-02003-f008:**
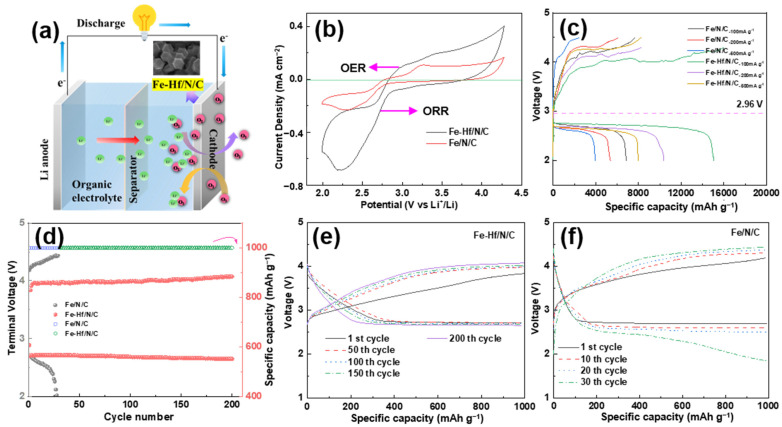
Charge–discharge performance and cycle stability: (**a**) A schematic diagram of the Li-O_2_ battery test device. (**b**) The CV curves of Fe-Hf/N/C and Fe/N/C at a scan rate of 0.3 mV s^−1^. (**c**) The rate capability profiles with different current densities. (**d**) The terminal voltage over a number of cycles. The voltage stability of (**e**) Fe-Hf/N/C and (**f**) Fe/N/C.

## Data Availability

Data will be made available on request.

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
