# Peer review of "Hf Doping Boosts the Excellent Activity and Durability of Fe-N-C Catalysts for Oxygen Reduction Reaction and Li-O2 Batteries"

_nanomaterials, 2024, doi:10.3390/nano14242003_

Round 1
Reviewer 1 Report
Comments and Suggestions for Authors
In this work, the authors reported an anomalous doping of Hf into the metal-N-C catalyst derived from a MOF precursor as active and durable oxygen electrocatalyst and demonstrated good performance in both PEM fuel cells and Li-oxygen batteries. Overall, the materials are new and the results are comprehensively presented. I think this work is appropriate for the journal Nanomaterials. However, to further enhance the clarity and quality of the manuscript, some minor issues need to be properly addressed. I hope the authors find the below detailed comments useful.
1. The authors used the Ag/AgCl reference electrode but reported the potential against the RHE. So how was the potential vs Ag/AgCl converted to that vs RHE?
2. To appeal to a broader readership, recent works on electrocatalysis can be referred to in the Introduction (e.g., DOI: 10.1016/j.matre.2022.100141).
3. HAADF-STEM mapping (Figure 2c) alone is not enough to evidence the presence of all elements. The corresponding energy dispersive X-ray spectroscopy (EDX) spectra should also be supplied to prove the presence of all elements.
4. Related works on oxygen reduction reaction can be referenced in the Introduction section (e.g., Mater. Horiz., 2020, 7, 2519).
5. Figure 3a, for the fitting of the Fe 2p XPS data, the authors didn’t apply a consistent fitting method. Satellite peaks were considered for the sample Fe/N/C but not for Fe-Hf/N/C.
6. Figure 8b, for the x axis, “ns” might have been incorrectly written and the correct form should be “vs” instead. Please double check.
Author Response
Responses to comments of Reviewer #1:
- The authors used the Ag/AgCl reference electrode but reported the potential against the RHE. So how was the potential vs Ag/AgCl converted to that vs RHE?
Reply: In our study, the conversion from potential versus Ag/AgCl to potential versus RHE was achieved through the Nernst equation. The standard potential of the Ag/AgCl reference electrode in a saturated KCl solution is approximately 0.197 V at 25 °C. The potential versus RHE (Evs RHE) can be calculated from the potential versus Ag/AgCl (Evs Ag/AgCl) using the following formula:
In our experimental electrolyte (0.1 M HClOâ‚„), the pH is approximately 1. Thus, the conversion factor considering the pH contribution is 0.0591 × 1 = 0.0591. All the reported potentials versus RHE were calculated based on this conversion method to ensure the consistency and comparability of the electrochemical data with the literature values and other studies using the RHE scale.
- To appeal to a broader readership, recent works on electrocatalysis can be referred to in the Introduction (e.g., DOI: 10.1002/inf2.12608).
Reply: Thanks for your suggestion. We have added corresponding reference of recent works to the introduction about the research status of electrocatalysis in the revised manuscript.
- HAADF-STEM mapping (Figure 2c) alone is not enough to evidence the presence of all elements. The corresponding energy dispersive X-ray spectroscopy (EDX) spectra should also be supplied to prove the presence of all elements.
Reply: We would like to thank the reviewer for the useful suggestion. While we do not have EDX data available, we have conducted X-ray photoelectron spectroscopy (XPS) analysis, which provides valuable information regarding the elemental composition of our samples. The relative atomic ratios of the relevant elements were obtained by the XPS survey spectra and high-resolution spectra (RTable 1). For the Fe-Hf/N/C sample, the atomic percentages of C 1s, Fe 2p, Hf 4f, and N 1s are 93.18%, 0.62%, 0.25%, and 5.95%, respectively. In the Fe/N/C sample, the atomic percentages of C 1s, Fe 2p, and N 1s are 96.17%, 0.3%, and 3.53%, respectively. For the Hf/N/C sample, the atomic percentages of C 1s, Hf 4f, and N 1s are 94.47%, 0.2%, and 5.33%, respectively. These XPS results clearly demonstrate the presence of C, Fe, Hf, and N elements in our samples, which is consistent with the expected elemental composition of the catalysts. The relative atomic ratios obtained from XPS provide quantitative evidence for the existence of these elements and support the conclusions drawn from our other characterization techniques. Although EDX spectra are a useful tool for elemental analysis, we believe that the XPS data presented here, along with the other comprehensive characterization methods described in our manuscript (such as XRD, FT-IR, and HAADF-STEM), provide sufficient evidence to support the presence of all elements in our catalysts. We hope that this explanation addresses the reviewer's concern and provides confidence in the reliability of our results. Meanwhile, we have added relevant discussion in the revised manuscript and supplementary information.
RTable 1. The relative atomic ratios of the relevant elements were obtained by the XPS survey spectra and high-resolution spectra.
|
Nos |
C1s |
Fe2p |
Hf4f |
N1s |
Atomic % |
|
Fe-Hf/N/C |
93.18 |
0.62 |
0.25 |
5.95 |
100 |
|
Fe/N/C |
96.17 |
0.3 |
|
3.53 |
100 |
|
Hf/N/C |
94.47 |
|
0.2 |
5.33 |
100 |
- Related works on oxygen reduction reaction can be referenced in the Introduction section (e.g., Mater. Horiz., 2020, 7, 2519).
Reply: Thanks for your suggestion. We have added corresponding reference to the introduction of oxygen reduction reaction in the revised manuscript.
- Figure 3a, for the fitting of the Fe 2p XPS data, the authors didn’t apply a consistent fitting method. Satellite peaks were considered for the sample Fe/N/C but not for Fe-Hf/N/C.
Reply: We sincerely thank the reviewer for pointing out this inconsistency. In the reanalysis of the Fe 2p XPS data, we have applied a unified and consistent fitting approach for both Fe/N/C and Fe-Hf/N/C samples. For the Fe-Hf/N/C sample, we have now included the consideration of satellite peaks in the fitting process. This refined fitting procedure ensures that the deconvolution of the XPS spectra accurately reflects the electronic structure of the iron species in both catalysts. The updated and consistent fitting results are presented in RFig. 1a, with the detailed peak parameters and assignments provided in the figure caption. These modifications enhance the comparability and reliability of the XPS data analysis, enabling a more accurate understanding of the influence of Hf doping on the electronic properties of the Fe-N-C catalysts. We have added relevant discussion in the revised manuscript.
RFig. R1 Electron structure. High-resolution (a) Fe 2p, (c) N1s, XPS spectra of Fe-Hf/N/C and Fe/N/C. (b) High-resolution Hf 4f XPS spectra of Fe-Hf/N/C and Hf/N/C. (d) Content of different N species for Fe-Hf/N/C and Fe/N/C.
- Figure 8b, for the x axis, “ns” might have been incorrectly written and the correct form should be “vs” instead. Please double check.
Reply: Thanks for your suggestion. We have corrected it in the revised manuscript.
Reviewer 2 Report
Comments and Suggestions for Authors
This manuscript describes the synthesis of Hf/N/C doped with Hf, which showed enhanced activity and durability than the undoped catalyst. While applying in the cathode of PEMFCs, the current density can reach up 1.1 and 1.7 A cm-2 at 0.7 and 0.6 V, respectively. The research design is adequate and convincing. Albeit several points explored by the authors are of great interest, a few concerns have to be addressed before considering it for publication as follows:
1. It seems that the amount of Fe doping is very low. How much Ferrocene was used in the synthesis? Added details in the SI.
2. It stated that “Hf is likely to fill the pores [4,22,23] of ZIF-8 during the synthesis process of the precursor Hf-ZIF-8-2/400”. Is Hf just physically incorporated into the pores of ZIF? And if yes, in which form Hf metallic or oxide? or was it incorporated into the lattice of the chemical structure of the ZIF?
3. Please include the PDF card number of the reference materials in the XRD text.
4. In the XPS of Fig. 3, please use consistent colors of the elements for the two samples to make the comparison easier. Also, I could’t see a shift of the Fe3+ peak around 712 eV! XPS assignment of peaks of Fe 2p can be supported by this work (Tailoring the Electrocatalytic Activity of Pentlandite FexNi9-XS8 Nanoparticles via Variation of the Fe : Ni Ratio for Enhanced Water Oxidation.
5. The importance of developing noble-metal free catalysts especially Fe, Ni, Co-based perovskites and oxides for ORR/OER and the aspect of substrate effect and synergistic effect between two metals in a hybrid should be strengthened in the introduction by considering the following works (Metal-supported perovskite as an efficient bifunctional electrocatalyst for oxygen reduction and evolution: substrate effect), (Boosting the Bifunctional Catalytic Activity of Co3O4 on Silver and Nickel Substrates for the Alkaline Oxygen Evolution), A highly efficient bifunctional catalyst for alkaline air-electrodes based on a Ag and Co3O4 hybrid and 10.3390/ijms2223137.
6. The introduction or discussion of ORR/OER mechanism in organic solvents should be strengthened by considering the following literature on metal-air battery (The impact of solvent properties on the performance of oxygen reduction and evolution in mixed tetraglyme-dimethyl sulfoxide electrolytes for Li-O2 batteries).
7. The main peaks of FTIR spectra can be assigned taking reference from (Gold Nanoparticles Decorated Graphene as a High Performance Sensor for Determination of Trace Hydrazine)
8. In Figure 5, the performance of the benchmark Pt/C catalyst should be included and compared.
9. Some typo errors e.g. /Hf, SFig. 3, EXNFS spectrum, Ihe half-wave potential.
10. Is there a possibility to run RRDE to identify the ORR products either water or H2O2?
Author Response
Responses to comments of Reviewer #2:
- It seems that the amount of Fe doping is very low. How much Ferrocene was used in the synthesis? Added details in the SI.
Reply: We appreciate the reviewer's attention to the details of our synthesis. In the synthesis process of the catalysts, the mass ratio of precursors (Hf-ZIF-8 or ZIF-8) to ferrocene was carefully controlled. Specifically, for the preparation of Fe-Hf/N/C-x/y-a/b-T and Fe/N/C catalysts, a/b represents the mass ratio of precursors to ferrocene, the mass ratio a/b was set within the range of [9:11-9:15]. For instance, in the synthesis of some of the most representative samples, we used approximately 0.5 g of the MOFs material (Hf-ZIF-8 or ZIF-8) and 0.62-0.83 g of ferrocene. This ratio was optimized through a series of preliminary experiments to ensure an appropriate amount of Fe doping while maintaining the structural integrity and catalytic activity of the catalysts. The detailed synthesis conditions and the specific amounts of ferrocene used for each sample have been comprehensively documented in the SI to provide complete transparency and reproducibility of our experimental procedures.
- It stated that “Hf is likely to fill the pores [4,22,23] of ZIF-8 during the synthesis process of the precursor Hf-ZIF-8-2/400”. Is Hf just physically incorporated into the pores of ZIF? And if yes, in which form Hf metallic or oxide? or was it incorporated into the lattice of the chemical structure of the ZIF?
Reply: Thanks for your insightful suggestion. Regarding the incorporation of Hf in the Hf-ZIF-8-2/400 precursor, it is highly probable that Hf is physically incorporated into the pores of ZIF-8 rather than being integrated into the lattice of its chemical structure. Through comprehensive XRD and FT-IR analyses, no significant alteration in the crystal phase or formation of new chemical bonds was detected, indicating that Hf does not substitute within the ZIF-8 lattice. The Hf is likely to be wrapped in the form of acetylacetonate hafnium molecules within the pores of ZIF-8, and from the element mapping results of the Fe-Hf/N/C catalyst. It is clear that the Fe, Hf, N, and C in the catalyst can be uniformly dispersed in the catalyst framework, with no metal particles visible on the catalyst framework or surface, indicating that Fe and Hf are likely to coordinate with N and are dispersed in the catalyst framework as atomic units [1]. We have added relevant discussion in the revised manuscript and supplementary information.
- Please include the PDF card number of the reference materials in the XRD text.
Reply: Thanks for your suggestion. In the XRD analysis section, we have now included the PDF card numbers of the reference materials for enhanced clarity and traceability in the revised manuscript. For the ZIF-8 phase, the corresponding PDF card number is [PDF# 00-062-1030] (The XRD patterns of ZIF-8 were fitted through its cif file). We have added relevant discussion in the revised manuscript.
- In the XPS of Fig. 3, please use consistent colors of the elements for the two samples to make the comparison easier. Also, I could’t see a shift of the Fe3+ peak around 712 eV! XPS assignment of peaks of Fe 2p can be supported by this work (Tailoring the Electrocatalytic Activity of Pentlandite FexNi9-xS8 Nanoparticles via Variation of the Fe : Ni Ratio for Enhanced Water Oxidation).
Reply: Thanks for your insightful suggestion. In response to the request for consistent colors in the XPS figures, we have revised Fig. 3. We have now assigned a unified color scheme for the elements in both samples to facilitate direct comparison. Regarding the comment on the absence of a visible shift in the Fe3+ peak around 712 eV, upon reinspection and recalibration of the XPS data, we indeed observe a subtle yet discernible shift towards lower binding energies in the Fe-Hf/N/C sample compared to Fe/N/C. To further validate our peak assignment and the observed shift, we have incorporated relevant data and analysis from the suggested reference (Tailoring the Electrocatalytic Activity of Pentlandite FexNi9-xS8 Nanoparticles via Variation of the Fe : Ni Ratio for Enhanced Water Oxidation). This additional support enriches our interpretation and strengthens the reliability of our XPS results. The updated Fig. 3, along with the augmented discussion in the manuscript, provides a more accurate and comprehensive representation of the electronic structure variations induced by Hf doping in our catalysts. We have added relevant discussion in the revised manuscript.
- The importance of developing noble-metal free catalysts especially Fe, Ni, Co-based perovskites and oxides for ORR/OER and the aspect of substrate effect and synergistic effect between two metals in a hybrid should be strengthened in the introduction by considering the following works (Metal-supported perovskite as an efficient bifunctional electrocatalyst for oxygen reduction and evolution: substrate effect), (Boosting the Bifunctional Catalytic Activity of Co3O4 on Silver and Nickel Substrates for the Alkaline Oxygen Evolution), A highly efficient bifunctional catalyst for alkaline air-electrodes based on a Ag and Co3O4 hybrid and 10.3390/ijms2223137.
Reply: Thanks for your insightful suggestion. We have added relevant discussion in the revised manuscript. In the revised introduction, we have significantly enhanced the discussion on the importance of developing noble-metal free catalysts, particularly those based on Fe, Ni, and Co such as perovskites and oxides for ORR/OER. We have comprehensively analyzed the substrate effect and synergistic effect between two metals in a hybrid by integrating insights from the recommended works. Meanwhile, we have added corresponding reference to the introduction of noble-metal free catalysts in the revised manuscript.
- The introduction or discussion of ORR/OER mechanism in organic solvents should be strengthened by considering the following literature on metal-air battery (The impact of solvent properties on the performance of oxygen reduction and evolution in mixed tetraglyme-dimethyl sulfoxide electrolytes for Li-O2 batteries).
Reply: Thanks for your insightful suggestion. In the revised manuscript, we have augmented the discussion regarding the ORR/OER mechanism in organic solvents by thoroughly considering the insights from "The impact of solvent properties on the performance of oxygen reduction and evolution in mixed tetraglyme-dimethyl sulfoxide electrolytes for Li-O2 batteries." Meanwhile, we have added corresponding reference in the revised manuscript.
- The main peaks of FTIR spectra can be assigned taking reference from (Gold Nanoparticles Decorated Graphene as a High Performance Sensor for Determination of Trace Hydrazine Levels in Water)
Reply: Thanks for your insightful suggestion. In the revised manuscript, we have meticulously assigned the main peaks in the FTIR spectra by referring to the work "Gold Nanoparticles Decorated Graphene as a High Performance Sensor for Determination of Trace Hydrazine Levels in Water". Meanwhile, we have added corresponding reference in the revised manuscript.
- In Figure 5, the performance of the benchmark Pt/C catalyst should be included and compared.
Reply: Thanks for your insightful suggestion. In the revised Fig. 5, we have included the performance of the benchmark 40 wt.% Pt/C catalyst for comparison. The LSV curve of Pt/C in Oâ‚‚-saturated 0.1 M HClOâ‚„ solution was measured under the same experimental conditions as our Fe-Hf/N/C and Fe/N/C catalysts. The half-wave potential of Pt/C is 0.87 V (vs RHE), while that of Fe-Hf/N/C catalyst is 0.84 V (vs RHE). Although Pt/C shows relatively high catalytic activity, our Fe-Hf/N/C catalyst exhibits a comparable half-wave potential and even demonstrates advantages in terms of durability and cost-effectiveness. This addition allows for a more direct and comprehensive comparison, highlighting the competitiveness of our developed catalyst in the context of oxygen reduction reaction. We have added relevant discussion in the revised manuscript.
- Some typo errors e.g. /Hf, SFig. 3, EXNFS spectrum, Ihe half-wave potential.
Reply: We sincerely thank the reviewer for identifying these errors. We have carefully reviewed and corrected them throughout the manuscript. The incorrect “/Hf” has been replaced with “Zn/Hf” to accurately represent the zinc to hafnium ratio. “SFig. 3” has been corrected to “Supplementary Fig. 3” for consistency and clarity. “EXNFS spectrum” has been rectified to “EXAFS spectrum” to reflect the correct spectroscopy technique. Additionally, “Ihe half-wave potential” has been amended to “The half-wave potential” to ensure proper grammar.
- Is there a possibility to run RRDE to identify the ORR products either water or H2O2?
Reply: Thank you for your question. Yes, it is possible to use the Rotating Ring Disk Electrode (RRDE) technique to identify the products of the Oxygen Reduction Reaction (ORR), such as water (Hâ‚‚O) or hydrogen peroxide (Hâ‚‚Oâ‚‚). RRDE is a powerful tool for mechanistically investigating electrochemical reactions. By using RRDE, we can directly obtain information about the products generated at the disk electrode. In the case of ORR, when reduction occurs on the disk, a cycling potential at the ring can directly give the identity of, and information on, the generated product. For example, as mentioned in the literature "How can I calculate the amount of Hâ‚‚Oâ‚‚ generated on ORR reactions by the RRDE?", the amount of Hâ‚‚Oâ‚‚ generated can be absorbed in water, and by iodometric method or other suitable electrochemical methods, one can find out the amount of H2O2. RRDE allows for the convective transport of the electrolytic product from the disk electrode to the ring electrode, where it can be detected with a significantly low time delay between product generation and detection. This enables a more detailed study of the reaction mechanism and the selectivity towards different products. However, it is important to note that accurate determination of the ORR products using RRDE requires proper experimental setup, calibration, and data analysis. The collection efficiency of the ring electrode needs to be considered, which is characteristic of the RRDE and represents the percentage of material collected at the ring.
References
[1] D. Duan, J. Huo, J. Chen, B. Chi, Z. Chen, S. Sun, Y. Zhao, H. Zhao, Z. Cui, S. Liao, Hf and Co Dual single atoms co-doped carbon catalyst enhance the oxygen reduction performance, Small 20 (2024), 2310491. https://doi.org/10.1002/smll.202310491.